# Income Change One Year after Confirmed Cancer Diagnosis and Its Associated Factors in Japanese Patients

**DOI:** 10.3390/ijerph192315992

**Published:** 2022-11-30

**Authors:** Akitsu Murakami, Kanae Kanda, Nlandu Roger Ngatu, Kosuke Chujo, Yusuke Yamadori, Yukinori Mashima, Akihito Tsuji, Tomohiro Hirao, Gotaro Shirakami

**Affiliations:** 1Department of Anesthesiology, Faculty of Medicine, Kagawa University, Miki-cho 761-0793, Kagawa, Japan; 2Department of Public Health, Faculty of Medicine, Kagawa University, Miki-cho 761-0793, Kagawa, Japan; 3Cancer Center, Kagawa University, Miki-cho 761-0793, Kagawa, Japan; 4Clinical Research Support Center, Kagawa University Hospital, Miki-cho 761-0793, Kagawa, Japan

**Keywords:** cancer survivor, continued employment, financial stress, income change, socioeconomic factors, year-on-year income level

## Abstract

The number of patients who survive for a long time after cancer diagnosis is rapidly increasing; however, such patients experience major problems such as returning to work and changes in their income. This study aimed to determine the extent of income changes of cancer patients during the first year after cancer diagnosis and identify the influencing factors. From November 2019 through January 2020, we conducted a multicenter, self-administered anonymous survey of cancer patients in Kagawa Prefecture, Japan. The number of questionnaires collected was 483 (recovery rate 60.4%), and the number of participants who met the inclusion criteria was 72. Mean year-on-year income level one year since cancer diagnosis was 66% (SD: 32%; median: 70%). Cancer stage (*p* = 0.016), employment status at diagnosis (*p* = 0.006), and continued employment at the same workplace (*p* = 0.001) were associated with income change. Findings from this study showed that cancer patients lost one-thirds of their income one year after their diagnosis. It was related to the stage of their illness, employment status, and continued employment at their workplace just before the diagnosis. Employers should provide cancer patients with the support they need to keep them employed.

## 1. Introduction

In recent years, owing to remarkable advances in cancer treatment, the relative 5-year survival rate of cancer patients has improved markedly to nearly 70% in developed countries [1,2,3,4]. The number of cancer patients who survive for a long time has increased considerably, and return to work (RTW) has become one of the ways for cancer survivors to rejoin society [5,6]. In Japan, the third basic plan to promote cancer control programs clearly states the necessity of support for balancing between treatment and employment [7]. Therefore, it is necessary to construct measures that support continued employment on the basis of specific surveys including employment status, but large-scale surveys in Japan have just begun [8].

With regard to employment support, the problem is that, in many cases, a decrease in income after cancer diagnosis has been observed [9,10]. Although studies on the change of income among cancer survivors exist, such as observational studies related to breast cancer, the magnitude of the income change has not been fully investigated [9,11,12,13,14,15]. In addition, the factors that influence the change have not been completely reported. Therefore, we set the research questions how the income would change in cancer survivors one year after the cancer diagnosis and what kind of factors contribute to the change. We hypothesized that job continuity after cancer diagnosis would influence income change in this study.

## 2. Materials and Methods

### 2.1. Target Population and Data Collection

We conducted this multicenter, cross-sectional questionnaire survey of cancer patients in Kagawa, Japan, from November 2019 through January 2020, using a self-administered anonymous questionnaire.

The participants were Japanese cancer patients attending or hospitalized at six major hospitals, including five regional and one pediatric cancer centers in Kagawa prefecture, which has a population estimated to be one million. For patients who had difficulty to fill in the questionnaire by themselves (elderly patients, children), answers were provided by their family members. Eight hundred copies of the questionnaire were distributed to the target institutions (150 copies to five cancer centers and 50 copies to the pediatric cancer center). The questionnaires were distributed directly to inpatients and outpatients in the order of their arrival at each facility and were collected in unmarked envelopes at each facility after the responses were completed. Among patients who consented to participate in this study, (1) only those whose age of cancer onset was 18 or older and (2) who have already spent one year or more after a definitive cancer diagnosis were included. The definition of “cancer survivor” has not been established [16]. In this study, patients who were diagnosed with cancer at least one year after the onset of symptoms were considered cancer survivors. The exclusion criteria were (1) cancer stage 0, (2) unknown cancer stage, (3) being unemployed at the time of diagnosis, and (4) unavailability of information required for the analysis.

### 2.2. Contents of the Questionnaire

For the change in personal income, cancer survivors were asked about the percentage change between just before and one year after their cancer diagnosis, i.e., their year-on-year income level (YIL [%]). They were asked to enter 100 if their income remained unchanged, a number greater than 100 when their income increased, and a number less than 100 when their income decreased.

We asked each respondent about the factors affecting RTW [7,9,11,12,17,18,19]: sex, age of onset, stage of cancer, treatment (surgery, radiation therapy, and pharmacotherapy), number of years since the diagnosis, support from family and friends, consultation with others, use of peer support, employment status at diagnosis, informing the workplace that you have cancer, support from the workplace, provision of information on continued employment via the medical staff, and continued employment. In the current study, a leave of absence was treated as a continuation of employment in the same workplace.

The variable “age of cancer onset” was divided into two categories: “60 years or older” and “younger than 60 years”. This is because most companies in Japan set the age of 60 years as the retirement age. Pharmacotherapy included chemotherapy, hormone therapy, molecular targeted therapy, and immunotherapy. Peer support was defined as patient support by cancer patients, cancer survivors, and their families, including patient groups. Patients who continued to work at the same workplace included respondents who were on leave.

### 2.3. Data Analysis

For the bivariate relationships between income change and each factor affecting RTW, the Wilcoxon test or Kruskal–Wallis test was performed. For multivariable analysis, analysis of covariance (ANCOVA) was used. JMP Pro 15.1 (SAS Institute Inc., Cary, NC, USA) was used as the statistical software. The sample size was set to detect an income difference of 20% for each factor. Although we did not have sufficient information, we assumed the standard deviation of income change to be 30, with 80% power, and a significance level of 5%. The calculated sample size for the two groups was 73. In this study, a statistically significant level was set at *p* < 0.05.

### 2.4. Ethics Statement

This study was approved by the Ethics Committee of Kagawa University Faculty of Medicine (approval no. 2019-104). Informed consent was explained in the explanatory document provided to each patient, and survey participation was confirmed by an item at the beginning of the questionnaire. The answers to all questions were voluntary and anonymous. Respondents were asked to mark their consent to their participation in the survey.

## 3. Results

The total number of questionnaires collected was 483 (recovery rate 60.4%). The number of participants who met the criteria was 72. Their characteristics are shown in Table 1. Males accounted for 58% of the total sample. The age of onset was less than 60 years in two-thirds of the patients. The tumor type was lung cancer in 17 patients (24%), gastric cancer in 8 patients (11%), colorectal cancer in 10 patients (14%), breast cancer in 11 patients (15%), and others in 26 patients (36%). Twenty-eight patients (39%) had stage I/II cancer, and 44 (61%) had stage III/IV cancer. Forty-six patients (64%) underwent surgery, 57 (79%) received pharmacotherapy, and 22 (31%) underwent radiotherapy. More than 90% had notified their employers that they had cancer, and 74% had received support from their workplace. However, less than 10% of the patients used peer support. At the time of diagnosis, 49 (68%) were working full-time, 13 (18%) were working part-time, and 10 (14%) were self-employed. Fifty-seven people (79%) remained in the same workplace.

The primary outcome, namely, YIL, compared with pre-diagnosis had a mean of 66% (standard deviation = 32%, median 70%) (Figure 1). Table 1 shows the bivariate relationship between income level one year after diagnosis and the related factors. In the analysis by age of onset, the YIL was lower in the younger age group: 50% in the group < 60 years compared with 74% in the group ≥ 60 years (*p* = 0.002). Regarding cancer stage, the decrease in income was greater for those in the advanced stage: 81% in group I/II vs. 57% in group III/IV (*p* < 0.001). There was also a significant difference in employment status at diagnosis (*p* = 0.017), i.e., 74% in the full-time group, 42% in the part-time group, and 57% in the self-employed group. The part-time group had the largest decrease in income. The decrease in income was larger without support from the workplace: 73% in the group with support from the workplace and 46% in the group without support from the workplace (*p* = 0.001). For those who continued to work at the same workplace, the difference was two-fold: 74% in those who continued to work at the same workplace and 37% in those who without (*p* < 0.001). Sex, age, treatment, consultation with others, and notification of the disease to the workplace were not significantly different.

Multivariable analysis (ANCOVA) was performed for the five items that showed significant differences: age of onset, stage, employment status at diagnosis, support from the workplace, and continued employment at the same workplace (Table 2). The analysis revealed that cancer stage (*p* = 0.016), employment status at diagnosis (*p* = 0.006), and continued employment at the same workplace (*p* = 0.001) were associated with YIL.

Table 3 shows the relationship between post-diagnosis employment continuity and employment status on income change. By considering that YIL was 100% before cancer diagnosis, the YIL was 80% for full-time workers who continued to work in the same workplace but was 53% for those who changed workplaces. Among part-time workers, the YIL of those in the same workplace was 55%, and those who changed workplaces had a YIL of 0%, i.e., no income.

## 4. Discussion

We found that the income of cancer survivors decreased by an average of 30% one year after cancer diagnosis in the Japanese environment. Given that this study was conducted at major hospitals covering an entire prefecture, the results were considered population-based characteristics.

The decrease in income after cancer diagnosis has been reported in other studies [14,15,20], and this was also the case in the current study. This study newly revealed that continued employment in the workplace affects income change. In terms of employment status at the time of diagnosis, full-time workers were the most likely to maintain their income level, whereas part-time workers had the largest negative impact on their income level. Although employment status is a factor that affects RTW [8,11], it was newly found to be also related to income change in the present study.

Table 3 shows the unique relationship between post-diagnosis employment continuity and employment status on income change. With regard to income one year after diagnosis, it is clear that those who continue to work at the same workplace are more likely to maintain their income regardless of their employment status. In other words, (1) when the participants continuously belonged to the same workplace, their income was more than half that of the previous year for all types of employment status: 80% for full-time, 55% for part-time, and 68% for those self-employed. (2) When participants did not continuously belong to the same workplace, their income decreased significantly: 53% for full-time, 0% for part-time, and 15% for self-employed. Regardless of whether participants continued to work at the same workplace, being a part-time worker had a greater impact on income than any other employment status. However, the negative impact on their income is enormous for those who resign or who work part-time after their diagnosis. Full-time workers who continue to work at the same workplace are the least likely to have their income affected one year after cancer diagnosis, whereas part-time workers who do not continue to work at the same workplace are the most likely to have their income reduced. Therefore, it is necessary to establish a public financial support system that is easy for part-time workers to use and to create an environment that makes it easy for them to be rehired in a stable work environment even if they stop working.

In this study, the association between support from the workplace and the income change was not significantly different in the multivariable analysis. However, a more detailed evaluation of support may reveal factors that can reduce the negative impact of cancer on income. It has been indicated that the impact of cancer on employment is highly individualized and includes not only medical factors but also personal factors, health status, psychosocial factors, work motivation, and workplace-related factors [21]. There is a need for further multifaceted evaluations of the impact on personal income.

In recent years, the financial toxicity of cancer patients, which is defined as the cause of their subjective financial distress together with objective financial burden such as changes in employment status, has been recognized. There are many factors related to financial toxicity, including sex (female related), low income, loss of income, younger age, adjuvant therapies, antineoplastic therapies, more recent diagnosis, advanced cancer, and no health insurance [21,22]. Financial toxicity is directly related to the well-being of cancer patients. The establishment of a support system that enables patients to maintain their income is very important to alleviate financial toxicity and to maintain their quality of life. To understand the current situation, a larger scale survey is needed.

We focused on income changes one year after diagnosis, but the next step is to evaluate the impact of a longer cancer survivor period. In addition, the disease characteristics of each cancer need to be considered. To date, most of the studies have been conducted on cancers with long-term survival, such as breast cancer. However, with the improvement in the life expectancy of cancer patients, it is necessary to conduct studies on various cancer types to establish specific support measures.

This study had several limitations. First, the response rate was low at 60%. In general, most reviewers demand a response rate of 70% or higher [23]. Although this does not guarantee a reduction in the risk of selective bias, we are required to improve the questionnaire by focusing on the careful selection of questionnaire items because the response rate is affected by the population and the nature of the questions. Second, in this study, we used a cross-sectional design, and there is a possibility that recall bias could occur, particularly as the years that have elapsed since the diagnosis increase. Third, the sample size was small, thus limiting the number of parameters for the multivariable analysis. Therefore, it was not possible to know in detail the influence of each factor on income change. This requires discretion in generalizing results. Fourth, the sample was skewed toward the elderly because all cancer patients were included in the study. There is no consensus on the age group that is susceptible to income changes. However, the income security systems for the elderly and non-elderly are very different because of several social security systems for the elderly in Japan, such as the retirement age system and the pension system. For example, the Japanese Government provides an old age pension for every citizen aged 65 years and above. Accumulating data on income changes in the non-elderly group, which has a large working population, is thought to be directly linked to solving modern problems that promote employment support for cancer survivors. Fifth, the scale of the company or business where the patient works is unknown. Finally, the health status of the patients at the time of the study was not considered. Given that performance status affects RTW [24], it will be necessary to also consider the relationship with income. In addition, less physical symptoms contribute to the promotion of RTW [24]; therefore, providing appropriate palliative care from an early stage may reduce income changes. Recently, the role of palliative care in cancer treatment has increased enormously. Early palliative care has been recommended, and the integration of palliative care and cancer treatment has been attempted [25].

A large-scale longitudinal study is needed to clarify the problem of income changes for cancer survivors including such perspectives in the future.

## 5. Conclusions

The income of cancer survivors one year after diagnosis was 66% of the previous year in Japan. Cancer stage, employment status at the time of diagnosis, and continued employment at the same workplace were associated factors. As a full-time member of the workplace where the patient was employed at the time of the cancer diagnosis, we believe that employment continuity after cancer diagnosis should be recommended to workers and employers.

## Figures and Tables

**Figure 1 ijerph-19-15992-f001:**
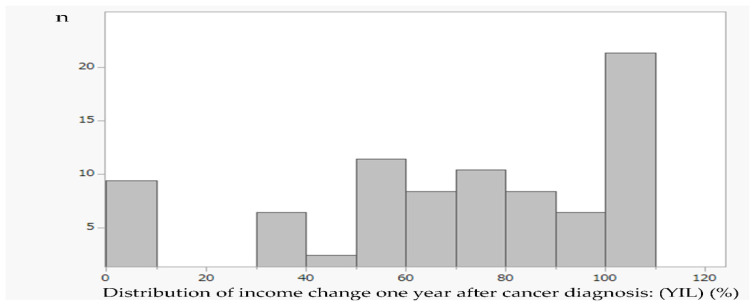
Distribution of income change one year after cancer diagnosis. Year-on-year income level (YIL) (%): the percentage change between just before and one year after their cancer diagnosis, *n* = 72. Mean (SD) = 66 (32)%, median = 70%. SD: standard deviation.

**Table 1 ijerph-19-15992-t001:** Characteristics of the participants (*n* = 72) and bivariate analysis results of the income change: YIL %(pre-diagnosis is set at 100).

Characteristics			Bivariate Analysis		
		*n* (%)	YIL Mean	SE	*p*-Value
Sex	Male	42 (58)	66	5.0	0.995
	Female	30 (42)	66	5.9	
Age of onset	<60 years	48 (67)	50	6.1	0.002
	≥60 years	24 (33)	74	4.3	
Years since onset	<5 years	49 (68)	61	4.5	0.057
	5 to <10 years	13 (18)	79	8.7	
	≥10 years	10 (14)	77	9.9	
Cancer site	Lung	17 (24)	63	7.2	0.176
	Gastric	8 (11)	77	9.4	
	Colorectal	10 (14)	72	8.3	
	Breast	11 (15)	72	6.8	
	Others	26 (36)	57	4.9	
Stage	I/II	28 (39)	81	5.7	0.000
	III/IV	44 (61)	57	4.5	
Surgery	Yes	46 (64)	69	4.7	0.174
	No	26 (36)	61	6.3	
Pharmacotherapy	Yes	57 (79)	67	4.3	0.989
	No	15 (21)	61	8.3	
Radiotherapy	Yes	22 (31)	66	6.9	0.829
	No	50 (69)	66	4.6	
Support from family and friends	Yes	68 (94)	66	3.9	0.479
	No	4 (6)	73	16.1	
Consultation with others	Yes	66 (92)	55	13.1	0.294
	No	6 (8)	67	4.0	
Sufficiency of the attending physician’s explanation	Yes	63 (88)	67	4.1	0.315
	No	9 (13)	60	10.7	
Using peer support	Yes	5 (7)	66	3.9	0.687
	No	67 (93)	62	14.4	
Employment status at diagnosis	Full-time	49 (68)	74	4.3	0.017
	Part-time	13 (18)	42	8.3	
	Self-employed	10 (14)	57	9.4	
Informing the workplace that you have cancer	Yes	69 (96)	66	3.9	0.460
	No	3 (4)	68	18.7	
Explanation of continuation of employment from medical staff	Yes	31 (43)	73	5.7	0.098
	No	41 (57)	61	5.0	
Support from employers/coworkers	Yes	53 (74)	73	4.1	0.001
	No	19 (26)	46	6.8	
Continued employment at the same workplace	Yes	57 (79)	74	3.8	0.000
	No	15 (21)	37	7.4	

YIL: year-on-year income level. SE: standard error.

**Table 2 ijerph-19-15992-t002:** Multivariate analysis of the income change: YIL % (pre-diagnosis is set at 100).

Variables			Multivariate Analysis		
			y-Intercept = 50.2		
		*n*	Estimate	SE	*p*-Value
Age of onset	<60 years	24	2.6	3.7	0.478
	≥60 years	48			
Stage	I/II	28	8	3.3	0.016
	III/IV	44			
Employment status at diagnosis	Full-time	49	12.5	4.7	0.006
	Part-time	13			
	Self-employed	10	1.9		
Support from employers/coworkers	Yes	53	6.3	3.5	0.078
	No	19			
Continued employment at the same workplace	Yes	57	13.1	3.9	0.001
	No	15			

SE: standard error.

**Table 3 ijerph-19-15992-t003:** Relationship between YIL by employment status and continued employment after diagnosis (pre-diagnosis is set at 100).

Employment Status at Diagnosis	Continued Employment at the Same Workplace					
	Yes			No		
	*n*	Mean	SE	*n*	Mean	SE
Full-time	39	80	4.3	10	53	6.6
Part-time	10	55	8.5	3	0	12.1
Self-employed	8	68	9.5	2	15	14.8

SE: standard error.

## Data Availability

The data presented in this study are openly available in Health and Welfare Department Health and Welfare General Affairs Division, Kagawa Prefecture at Cancer information, https://www.pref.kagawa.lg.jp/kenkosomu/yobou/cancer/kfvn.html (accessed on 10 October 2022), reference number [https://www.pref.kagawa.lg.jp/documents/3585/s77o6w171019144934_f72.pdf (accessed on 10 October 2022)]. (in Japanese).

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
