# Peer review of "Income Change One Year after Confirmed Cancer Diagnosis and Its Associated Factors in Japanese Patients"

_ijerph, 2022, doi:10.3390/ijerph192315992_

Round 1

Reviewer 1 Report

This is a very original and interesting paper dealing with a relevant emerging social  issue. Despite the small sample size and some other limitations (thoroughly acknowledged by the Authors themselves in the Discussion at page 8), I do think that this might be considered as a pilot study that leads the way to further epidemiological studies, which should focus not only on cancer patients but also on patients affected by other cronic diseases with negative effects on work ability and employability.

Given the above, minor text editing is required as follows:

1. affiliations, line 8, "Department of Public Health" repeated twice;

2. Keywords, line 29, "socinoeconomic" instead of socioeconomic;

3. Data Analysis, line 98, "the total sample size of the two groups is 73", but in Results, line 109 "The number of participants...was 72" and also in tables the reported number of participants is 72. Then, 73 or 72 effective participants?

4. Results, page 3, there is a repetition: lines 115-117 "More than 90% had notified their employers that they had cancer, and 74% had received support from their workplace"; lines 119-120 "More than 90% had notified their employers that they had cancer, and 74% had received support from their workplace".

5. Table 1, page 4: Age of onset, "60 years"...>= is missing.

6. References, line 268, before 2013 ( is missing.

7. References, line 274, "accessed on 26 August 2021"...I think that is 2022 instead.

Author Response

Dear Reviewer:

We are very grateful for the accurate comments on our paper.

We believe that the quality of the paper has been improved by pointing out problems that we were not aware of, and by making corrections and additions to each item according to the suggestions.

We thank you for your careful review of our paper.

Sincerely,

Akitsu Murakami

Tomohiro Hirao

Reviewer 2 Report

Authors have performed an interesting research about the income change one year after confirmed cancer diagnosis in Japanese Patients. The introduction is clear, concise, and help the reader understand the context of the question they want to answer with this study. This section could be enhanced by writing the hypothesis authors have in mind about this topic.

Methods are clearly described, but I would recommend including numbers (1) before each inclusion and exclusion criteria to ease the reader see them.  

In the results, I think there is a mistake on the first paragraph, as the same information is written twice “more than 90 % had notified their employers that they had cancer, and 74 % had received support from their workplace” please revise this section and delete the duplicated information. Also in the tables, please indicate those results that are statistically significant, again to ease the reader understanding these results.

Figure 1 is not really clear. Please include some more information to explain the data on the Y and the X coordinate axis.  

Same on the tables, it does not make sense that you calculate the mean on a categorical variable as sex (male, female). What does this information provide to the reader? Please check carefully this information, and perform the necessary changes in the results and the methods; categorical variables are not analyzed as continuous variables. Categorical variables may be exposed as (n) and percentages. Please modify this on all categorical variables and clarify the information on the tables.

Discussion is well written and understandable, but I guess when modifying the results, discussion must be also modified. Limitations of the study are well described.

In the conclusion, please be careful when writing your results. Your sample size is small, so be careful readers do not overestimate your results.

Author Response

(The authors gave the same response as above.)

Round 2

Reviewer 2 Report

Authors have adequatly modified their manuscript in order to improve it. Now paper is enhanced, and it can be published.